# Genome-Wide and Expression Pattern Analysis of the DVL Gene Family Reveals *GhM_A05G1032* Is Involved in Fuzz Development in *G. hirsutum*

**DOI:** 10.3390/ijms25021346

**Published:** 2024-01-22

**Authors:** Yang Jiao, Fuxiang Zhao, Shiwei Geng, Shengmei Li, Zhanlian Su, Quanjia Chen, Yu Yu, Yanying Qu

**Affiliations:** 1Cotton Research Institute, Xinjiang Academy of Agriculture and Reclamation Science, Shihezi 832000, China; jycotton@163.com (Y.J.); 18290810845@163.com (F.Z.); 2College of Agriculture, Xinjiang Agricultural University, Urumqi 830052, China; gengshiwei20201231@163.com (S.G.); lishengmei20201231@163.com (S.L.); 17799668420@163.com (Z.S.); chqjia@126.com (Q.C.)

**Keywords:** cotton, DVL, expression pattern, fuzz, WGCNA, *GhM_A05G1032*, VIGS

## Abstract

DVL is one of the small polypeptides which plays an important role in regulating plant growth and development, tissue differentiation, and organ formation in the process of coping with stress conditions. So far, there has been no comprehensive analysis of the expression profile and function of the cotton *DVL* gene. According to previous studies, a candidate gene related to the development of fuzz was screened, belonging to the DVL family, and was related to the development of trichomes in *Arabidopsis thaliana*. However, the comprehensive identification and systematic analysis of DVL in cotton have not been conducted. In this study, we employed bioinformatics approaches to conduct a novel analysis of the structural characteristics, phylogenetic tree, gene structure, expression pattern, evolutionary relationship, and selective pressure of the DVL gene family members in four cotton species. A total of 117 *DVL* genes were identified, including 39 members in *G. hirsutum*. Based on the phylogenetic analysis, the DVL protein sequences were categorized into five distinct subfamilies. Additionally, we successfully mapped these genes onto chromosomes and visually represented their gene structure information. Furthermore, we predicted the presence of *cis*-acting elements in *DVL* genes in *G. hirsutum* and characterized the repeat types of *DVL* genes in the four cotton species. Moreover, we computed the Ka/Ks ratio of homologous genes across the four cotton species and elucidated the selective pressure acting on these homologous genes. In addition, we described the expression patterns of the DVL gene family using RNA-seq data, verified the correlation between *GhMDVL3* and fuzz development through VIGS technology, and found that some *DVL* genes may be involved in resistance to biotic and abiotic stress conditions through qRT-PCR technology. Furthermore, a potential interaction network was constructed by WGCNA, and our findings demonstrated the potential of *GhM_A05G1032* to interact with numerous genes, thereby playing a crucial role in regulating fuzz development. This research significantly contributed to the comprehension of *DVL* genes in upland cotton, thereby establishing a solid basis for future investigations into the functional aspects of *DVL* genes in cotton.

## 1. Introduction

Cotton, the most important natural fiber crop, provides raw materials for the textile industry and plays a significant role in modern economic activities [1,2]. The growth and development of its internal ovules produce cotton fiber. The developmental stages of cotton fiber include four phases: fiber differentiation, primary wall formation, secondary wall synthesis, and others. Each stage overlaps without distinct time boundaries [3,4]. The initial and early elongation stages of cotton fiber are crucial and highly relevant to cotton yield and fiber quality, such as fiber density, length, and uniformity [5,6]. However, environmental factors often impact these stages and subsequently affect cotton fiber quality [7]. Therefore, it is essential to improve cotton fiber quality through molecular breeding.

Cotton fiber can be categorized into two types: fiber, which typically forms before or on the day of flowering, and fuzz, which generally forms 3–5 DPA (days post-anthesis) after flowering [8]. Previous research has primarily focused on fiber development, with less attention given to fuzz. Investigating the initiation of fuzz is crucial for comprehending the developmental mechanism of cotton fibers. Additionally, during cottonseed production, strong acid is employed to remove the fuzz from seed surfaces, resulting in increased costs and environmental pollution. Consequently, fuzzless characteristics are increasingly recognized as environmentally beneficial. With advancements in cotton genomics, numerous genes associated with fuzz development have been identified. In, *G. hirsutum*, *GhMML3_A12* is considered a potential gene linked to fuzz development [9]. Similarly, in diploid cotton research, *GaFzl* is deemed a candidate gene associated with the fuzzless trait of *G. arboreum* [10].

The exploration and functional analysis of plant peptides are emerging fields of research, especially in the context of responding to unfavorable environmental stress conditions. They play a significant role in regulating plant growth, development, tissue differentiation, organ formation, and other related processes. Small peptides typically possess a signal peptide at their N-terminus and a structural domain at their C-terminus. Generally, their amino acid composition does not exceed 150. Among these, DVL is a type of small peptide involved in regulating normal plant development. Previous studies have demonstrated that overexpressing this gene in *Arabidopsis* affects the development of trichomes [11], similar to the mechanism of cotton fiber development [12]. Moreover, as research progresses, an increasing number of small peptides are being identified and discovered. Under adverse environmental stress conditions, small peptides can effectively enhance plant resistance. For instance, RALF small peptides have been found to alkalinize tissue cell walls in *Arabidopsis* and rice roots, thereby inhibiting tissue growth and extension to regulate growth in response to adverse environmental stress and promote vitality. In terms of stomata regulation in plants, researchers have discovered that CLE25 small peptides indirectly control stomatal closure and expansion by binding to the BAM receptor in leaf tissue, thereby influencing leaf transpiration [13].

In this investigation, we identified a putative gene (*GhMDVL3*) linked to the development of fuzz based on previous findings from BSA-seq and RNA-seq analyses (unpublished). We performed a comprehensive exploration and examination of the DVL gene family members in four cotton species (*G. arboreum*, *G. raimondii*, *G. hirsutum*, and *G. barbadense*) utilizing diverse bioinformatics strategies. These approaches encompassed phylogenetic analysis, gene structure analysis, the identification of conserved motifs, and sequence analysis. The primary goal of these analyses was to ascertain the evolutionary relationships among *DVL* genes in cotton. Furthermore, we conducted a collinearity analysis to assess the Ka/Ks substitution ratio (Ka/Ks ratio). Additionally, we investigated the expression patterns of *GhMDVL* genes through promoter *cis*-element analysis and tissue-specific expression analysis. Finally, we performed the initial functional validation of the *GhMDVL3* gene using Virus-Induced Gene Silencing (VIGS) technology. This research provides innovative insights into the functional genomics of cotton and establishes a foundation for further investigations into the mechanisms underlying cotton fiber development.

## 2. Results

### 2.1. Genomic Identification of DVL Gene

We employed the local BLASTP program to perform a comparative analysis and search for potential sequences in the target files. Through this procedure, we acquired candidate sequences. Next, we verified the presence of the conservative domain using Pfam, SMART, and CDD. Any sequences lacking the complete DVL domain were discarded. Ultimately, we identified a total of 117 *DVL* genes across four cotton species (Appendix A). These species include *G. arboreum* (A2) with 20 genes, *G. raimondii* (D5) with 19 genes, *G. hirsutum* (AD1) with 39 genes, and *G. barbadense* (AD2) with 39 genes.

Significantly, the quantity of *G. raimondii* (D5) was observed to be one unit lower in comparison to *G. arboreum* (A2). Similarly, the total number of DVL family members in the two tetraploid cotton species precisely equals the sum of the quantities of the two diploid cotton species.

These genes were renamed *GaDVL1-GaDVL20*, *GbMDVL1-GbMDVL39*, *GhMDVL1-GhMDVL39*, and *GrDVL1-GrDVL19* (Appendix A) based on their location on the chromosome. Subsequently, the amino acid sequences of four cotton species belonging to the DVL gene family were subjected to analysis. The length of the amino acids within the DVL family members varied from 50 to 157 residues, with an average sequence length of 74 amino acids. The molecular weight ranged from 5.88 to 18 kDa, with an average of 8.59 kDa. The isoelectric point (pI) spanned from 5.89 to 11.66, with an average value of 9.78 (Appendix A).

### 2.2. Phylogenetic Analysis of DVL Gene

In order to investigate the evolutionary relationships among members of the DVL family, we employed MEGA software v7.0 to construct a phylogenetic tree. This tree was constructed using a dataset consisting of 117 protein sequences obtained from four different cotton species (Figure 1). Subsequently, the 117 DVL family members were categorized into five distinct subfamilies, designated as Cluster1–Cluster5. Among these clusters, Cluster5 exhibited the highest count of 28, followed by Cluster4 and Cluster2 with 26 and 24, respectively, and Cluster1 with 19 members. Notably, excluding Cluster4, the remaining four subfamilies had an equal distribution of *G. arboreum* and *G. raimondii*. In Cluster1, the count of *G. hirsutum* exceeded that of *G. barbadense* by one, whereas in Cluster4, it was the opposite. It is of significance to emphasize that the diploid species *G. arboreum* and *G. raimondii* consistently formed clusters together with the tetraploid species *G. hirsutum* and *G. barbadense*. This observation provides support for the hypothesis that *G. hirsutum* and *G. barbadense* originated from a hybridization event involving *G. arboreum* and *G. raimondii* [14].

### 2.3. The Chromosomal Mapping of DVL Genes in Four Cotton Species

In order to obtain a comprehensive understanding of the chromosomal distribution and gene duplication patterns of *DVL* genes in four different cotton species, a mapping analysis was performed to precisely determine the precise physical locations of these genes on the respective chromosomes. The 117 *DVL* genes were found to be randomly dispersed across the chromosomes of the four cotton species (Figure 2). In, *G. hirsutum*, 39 genes were observed to be randomly distributed among 16 chromosomes, with the number of *DVL* genes on each chromosome varying from one to seven. Notably, the A05 chromosome harbored the highest count of genes, with seven present. Among the 39 genes in *G. hirsutum*, 20 were categorized under subgenome A, while 19 were assigned to subgenome D. Subgroup A contained one additional gene compared to subgroup D. However, the following chromosomes did not exhibit any distribution of *DVL* genes: A01, A02, A03, A08, A12, D01, D02, D03, D08, and D12. The distribution pattern of *DVL* genes in *G. barbadense* mirrored that of *G. hirsutum*, with 39 genes randomly dispersed across 16 chromosomes, featuring a range of one to seven genes per chromosome. Similarly, the following chromosomes lacked *DVL* gene distribution: A01, A02, A03, A08, A12, D01, D02, D03, D08, and D12. In, *G. arboreum*, 20 *DVL* genes were observed across eight chromosomes, excluding Chr01, Chr02, Chr03, Chr08, and Chr12. The lowest gene count was found on Chr11, which contained only one gene, while the highest gene count was observed on Chr05, housing five genes. Likewise, in *G. raimondii*, 19 *DVL* genes were distributed across seven chromosomes, excluding Chr01, Chr02, Chr03, Chr08, Chr11, and Chr12. Unlike, *G. arboreum*, Chr11 did not exhibit any distribution of *DVL* genes.

### 2.4. Gene Structure and Protein Motif Analysis

To acquire a more comprehensive comprehension of the potential structural evolutionary connections among members of the DVL family in *G. hirsutum*, we employed the NJ method to build phylogenetic trees for the 39 *DVL* genes in *G. hirsutum*. Furthermore, motif association analysis and gene structure analysis were conducted to examine and evaluate the relationship between motifs and gene structures (Figure 3). By employing the protein sequences and annotation files of the 39 DVL members, we generated the phylogenetic tree and obtained gene structure information. To identify the conserved motifs in DVL proteins, we employed MEME and TBtools-II, eventually discovering 10 motifs among the 39 cotton members. The number of motifs differed among the members within each group, while the motif composition remained relatively similar across the groups. Nearly all *DVL* genes contained motif1, except for *GhMDVL18* and *GhMDVL38*. Subsequently, motif3 and motif4 emerged. Among these, *GhMDVL9* and *GhMDVL29* from group 3 exhibited the highest number of motifs, with five distinct motifs. It is noteworthy that only motif1 was present in *GhMDVL10* and *GhMDVL30* from Group 1, as well as *GhMDVL5* and *GhMDVL15* from Group 3, suggesting potential evolutionary mutations. To gain deeper insights into the gene structure of DVL family members in *G. hirsutum*, an investigation was conducted to analyze the specific features pertaining to the arrangement of introns and exons. As depicted in Figure 3A, members within the same group displayed similar intron–exon arrangements. Notably, *GhMDVL23* from group 1 exhibited two exons. Within group 3, *GhMDVL9*, *GhMDVL16*, *GhMDVL29*, and *GhMDVL35* possessed two exons, while *GhMDVL15* had three exons, warranting attention. Finally, group 3 exhibited multiple introns.

Simultaneously, a comprehensive analysis was performed on 20 *DVL* genes retrieved from *G. arboreum*, encompassing motif association analysis and gene structure analysis (Figure 3B). Among these genes, motif1 was found in all 19 genes, with the exception of *GaDVL9*. Notably, *GaDVL4*, *GaDVL10*, and *GaDVL19* exclusively possessed motif1, suggesting its high conservation among *G. arboreum*. Furthermore, in the examination of the gene structure, only *GaDVL1* and *GaDVL11* exhibited a single intron.

### 2.5. Cis-Regulatory Element Analysis of DVL Genes

To enhance our understanding of the regulatory mechanism of *DVL* genes, we employed the PlantCARE database to anticipate the *cis*-acting elements present in the 2000 bp promoter region located upstream of 39 *DVL* genes in *G. hirsutum* and 20 *DVL* genes in *G. arboreum*. In, *G. hirsutum* (Figure 4A), which includes *cis*-acting elements common in promoter and enhancer regions, *cis*-acting elements that are involved in the induction of the drought response often contain MYB binding sites. Other *cis*-acting elements associated with plant hormones include abscisic acid response elements, salicylic acid response elements, MeJA response elements, and auxin-responsive elements. Additionally, there are regulatory elements that are specifically involved in seed-specific regulation: the light-responsive element, the regulatory element involved in zein metabolism regulation, and the element involved in defense and stress responsiveness. It is worth noting that in *G. arboreum* (Figure 4B), *GaDVL7*, *GaDVL17,* and *GaDVL20* contain the fewest types of *cis*-acting elements, while in *G. hirsutum*, *GhMDVL5* and *GhMDVL8* contain the fewest types of *cis*-acting elements. Most members of the *DVL* gene contain light responsive elements, except *GhMDVL5*, *GhMDVL6*, *GhMDVL25*, *GhMDVL28*, *GaDVL5*, *GaDVL10*, and *GaDVL17*. By examining the promoter, we can validate the subsequent gene function.

### 2.6. Tissue Expression Patterns of DVL Genes in G. hirsutum

Gene expression patterns are closely related to their functions. To investigate the changes in the gene expression levels of DVL family members in different tissues, we first analyzed transcriptome data during ovule and fiber development [15]. In the whole development period of fiber and ovule, the expression levels of 39 *DVL* genes in the fiber development period were lower than those in the ovule development period. *GhMDVL14* and *GhMDVL34* had higher expression levels than other genes in six stages of ovule development (−3, −1, 0, 1, 3, and 5 DPA (days post-anthesis)). In addition, the 3 DPA expression of *GhMDVL19* was higher than that of *GhMDVL14* and *GhMDVL34* before flowering. Interestingly, we found that 39 *DVL* genes had lower expression levels in the later stages of ovule development (10, 20, 25, and 35 DPA), while *DVL* genes had higher expression levels in the early stages of ovule development, suggesting that *DVL* genes might play an important role in the early stages of fiber development (Figure 5A). Then, we compared the expression levels of the fuzz material Xinluzao 50 (fz) and the fuzzless material Xinluzao 50 FLM (FZ) (Figure 5B), and we found that the expression levels of *GhMDVL14* and *GhMDVL34* in the two materials were indeed higher than the expression levels of other genes in the early stage of fiber development. The expression levels of *GhMDVL14* and *GhMDVL34* on the first day after flowering were higher in fuzzless materials than in fuzz materials, and their expression levels were 2.3 and 2.2 times of those in fuzz materials, respectively. These two genes may negatively regulate fiber development in 1 DPA [16].

To investigate the expression level of the *DVL* gene in the development of fibers and ovules, we also studied its expression changes in other tissues (Figure 5C). The expression levels of *GhMDVL14*, *GhMDVL17*, *GhMDVL34*, *GhMDVL19,* and *GhMDVL39* in Pistil were higher than those of other genes. The expression levels of *GhMDVL2*, *GhMDVL16*, *GhMDVL35*, *GhMDVL19,* and *GhMDVL39* in Calycle were higher than those of other genes. Interestingly, *GhMDVL34* exhibited a higher expression level in the Pistil compared to *GhMDVL35*. In contrast, *GhMDVL35* demonstrated a higher expression level in the Calycle in comparison to *GhMDVL34*. Furthermore, *GhMDVL2*, *GhMDVL19*, and *GhMDVL39* displayed higher expression levels in the root compared to the other genes [15].

The expression levels of *GhMDVL34*, *GhMDVL14* and *GhMDVL2* were higher than those of other genes in the three stages of seed development, and the expression levels of *GhMDVL34* and *GhMDVL14* were the highest at 5 h after the seed germination stage (Figure 5D). The expression level of *GhMDVL16* at 0 h of seed germination was higher than that at other stages, indicating that the gene was specifically expressed at this stage. During root and cotyledon development, the expression of *GhMDVL13* at 96 h of cotyledon development was higher than that at other stages, indicating that this gene was specifically expressed at this stage. In addition, we found that the expression of *GhMDVL13* began to increase at 14 h of cotyledon development, reached its peak at 96 h, and then decreased at 120 h. Compared with the changes in the expression levels of cotyledon, there was no significant difference in the expression levels of *DVL* genes during root development, and *GhMDVL34*, *GhMDVL25,* and *GhMDVL3* were higher than those of other genes during five stages of root development [15].

Furthermore, we investigated the expression patterns of *DVL* genes in the pigment glands of cotton (as shown in Figure 5E), which have limited utility due to the presence of toxic gossypol within these tissues [17]. Consequently, studying the development of pigment glands becomes crucial for maximizing the utilization of cotton seeds. By analyzing the transcriptome data of true leaves from glanded varieties (L7 and Z17) and glandless varieties (L7XW and Z17YW), we observed that the expression of *GhMDVL16* was relatively high in all four materials. Interestingly, *GhMDVL31* and *GhMDVL35* were significantly different compared with other genes in glanded and glandless materials. The expression of *GhMDVL31* in Z17YW exhibited a higher level compared to Z17, with a 1.73-fold increase. This finding suggests that the gene potentially exerts a negative regulatory effect on the development of pigment glands in upland cotton. Similarly, the expression of *GhMDVL35* in L7XW displayed a higher level than in L7, with a 2.3-fold increase. These results imply that the gene may also negatively regulate the development of pigment glands in upland cotton.

### 2.7. Expression Patterns of DVL Genes in G. hirsutum under Stress Conditions

In order to elucidate the molecular mechanisms underlying the response of *DVL* genes to abiotic stress, we employed RNA-seq data to assess alterations in gene expression patterns in response to cold, heat, salt, and PEG-induced stress conditions (Figure 6A). The results showed that the expression levels of *GhMDVL35* and *GhMDVL16* were higher than those of other family members under the four abiotic stress conditions, indicating that these two genes were involved in the regulation of abiotic stress. The expression of *GhMDVL35* at 3 h was higher than that at other periods under hot stress. Under cold stress, the expression level of 3 h began to increase, reached the peak at 6 h, and then decreased at 12 h. Under salt stress, the expression level of 1 h was higher than that of other periods. Similarly, under PEG stress, the expression level of 1 h was higher than that of other periods. The expression levels of *GhMDVL16* were observed to be elevated at 0 h in response to hot stress, surpassing those seen at other time points. Conversely, under cold stress, the expression level was lower at 1 h compared to the initial 0 h time point. Subsequently, there was a gradual increase in expression at 3 h, reaching a peak at 6 h, followed by a subsequent decline. Under salt stress, with the increase of stress time, the expression level showed a trend of gradual decline. The same phenomenon also appears under PEG stress [15].

### 2.8. Expression Analysis of the DVL Genes in G. hirsutum in Response to Verticillium Wilt Stress

*Verticillium wilt* is a devastating plant disease which has seriously affected cotton production [18]. The present investigation utilized transcriptome data from cotton roots following inoculation with *Verticillium dahliae*. The results showed (Figure 6B) that the expression levels of *GhMDVL38* and *GhMDVL18* genes decreased at 0 h, 6 h, and 12 h after infection by *Verticillium dahliae*. The expression levels began to increase 24 h later, and then decreased with the increase in the infection time. These two genes have the potential to participate in the cotton plant’s response to *Verticillium dahliae* treatment. The gene *GhMDVL17* exhibited an initial increase in expression at 6 h, followed by a subsequent decrease at 12 h after infection with *Verticillium dahliae*. This suggests that *GhMDVL17* may play a role in the early response of cotton to treatment with *Verticillium dahliae*.

### 2.9. Expression Analysis of the DVL Genes in G. hirsutum in Response to TDZ Treatment

Mechanized harvesting is the current direction of cotton production in China, serving as a crucial solution to enhance efficiency and address labor shortages. The pre-harvest application of chemical defoliant aids in inducing leaf shedding, facilitating the opening of cotton bolls, significantly reducing impurity levels in cotton bolls, and ultimately enhancing the effectiveness of mechanized harvesting. Thidiazuron (TDZ), a synthetic compound with cytokinin-like properties, is a key ingredient in chemical defoliants used in cotton production. Nevertheless, the effectiveness of TDZ is strongly influenced by environmental factors, particularly temperature. The present investigation utilized transcriptome data obtained from TDZ-sensitive cotton varieties, specifically CIR12 and CCIR50 [19]. The results showed that the expression level of *GhMDVL2* at 24 h after TDZ treatment was 50 times that of the control group and 75 times that of the control group at 48 h. In addition to *GhMDVL2*, we found that *GhMDVL24* is similar to *GhMDVL2*, indicating that these two genes could potentially play a role in the cotton’s reaction to TDZ treatment. Interestingly, we also found that the expression level of *GhMDVL2* after 144 h of TDZ treatment at 15 °C (a low temperature) was 117 times that under 25 °C (a normal temperature), indicating that this effect increased with time after TDZ treatment at a low temperature. In addition, it was observed that *GhMDVL17* exhibits a similarity to *GhMDVL2* and may also contribute to the cotton’s response to TDZ treatment under low temperature conditions (Figure 6C).

### 2.10. Expression Analysis of the DVL Genes in G. barbadense in Response to FOV Stress

*Fusarium oxysporum f.* sp. *vasinfectum* (FOV) is a pathogenic fungus that resides in the soil and poses a significant threat to cotton cultivation, particularly in *G. barbadense* varieties [20]. This study was based on the transcriptome data of cotton roots 40 h after inoculation with *Fusarium oxysporum* (Figure 6D). The results showed that 90% of *GbMDVL* gene expression has no obvious change before and after the infection, but *GbMDVL2*, *GbMDVL6*, *GbMDVL18*, and *GbMDVL23* may play an important role in the FOV resistance of *G. barbadense*.

### 2.11. Gene Replication and Collinearity Analysis

The evolution of gene families primarily involves whole genome duplication, segmental duplication, and tandem duplication. In order to explore the evolutionary processes and impact of polyploidy, we conducted a comprehensive analysis to identify the repetitive patterns of *DVL* genes in four different cotton species (Appendix A). Among the two diploid cotton species, 20 *DVL* genes of *G. arboreum* and 19 *DVL* genes of *G. raimondii* belong to WGD or Segmental. However, in tetraploid cotton, most of the 39 *DVL* genes of *G. hirsutum* belonged to WGD or Segmental, among which one gene belonged to Tandem and one gene belonged to Dispersed. *G. barbadense* was similar to *G. hirsutum*, and most of the 39 *DVL* genes belonged to WGD or Segmental. Only one gene was Dispersed.

We conducted a comprehensive multicollinearity analysis of *DVL* genes in four cotton species, namely *G. hirsutum* (AD1), *G. barbadense* (AD2), *G. arboreum* (A2), and *G. raimondii* (D5) (Figure 7A). Our analysis revealed that *G. hirsutum* and *G. arboreum* exhibited 89 orthologous gene pairs, *G. barbadense* and *G. arboreum* exhibited 91 orthologous gene pairs, *G. barbadense* and *G. barbadense* exhibited 61 orthologous gene pairs, *G. barbadense* and *G. hirsutum* exhibited 76 orthologous gene pairs, *G. barbadense* and *G. raimondii* exhibited 92 orthologous gene pairs, *G. hirsutum* and *G. raimondii* exhibited 92 orthologous gene pairs, and *G. raimondii* and *G. raimondii* exhibited 16 orthologous gene pairs. Based on these findings, we postulate that the primary drivers of gene amplification in the evolutionary process of *DVL* family genes are genome-wide replication events and fragment replication events.

Furthermore, we conducted a collinearity analysis between subgroup A and subgroup D of tetraploid *G. hirsutum* (Figure 7B), identifying 58 orthologous/paralogous pairs, and also analyzed the collinearity analysis between *G. arboreum* genomes (Figure 7C). A total of 13 orthologous/paralogous pairs were found.

### 2.12. Calculation of Selection Pressure

To investigate the mechanisms underlying the differentiation of *DVL* genes during polyploid replication events in cotton, we calculated the ratios of non-synonymous to synonymous substitutions (Ka/Ks ratios) to assess the selection pressures acting on these homologous gene pairs during evolution (Appendix A). A total of 588 homologous gene pairs were analyzed across four cotton species. The Ka/Ks ratios between *G. arboreum* and *G. raimondii* were all below 0.5, as were the ratios between *G. raimondii* and *G. raimondii*. However, among tetraploid *G. hirsutum*, 49 homologous gene pairs exhibited Ka/Ks ratios below 0.5, while three pairs had values between 0.5 and 0.99. Similarly, in *G. barbadense*, 51 homologous gene pairs had Ka/Ks ratios below 0.5, with three pairs having values between 0.5 and 0.99. These findings suggest that these *DVL* genes have undergone strong purifying selection during their evolutionary history. However, one homologous gene pair between *G. hirsutum* and two homologous gene pairs between *G. barbadense* displayed Ka/Ks ratios greater than one, indicating a positive selection and suggesting the recent rapid evolution of these genes, which may hold important implications for species evolution (Figure 7D).

Furthermore, we also calculated the Ka/Ks ratios between 89 pairs of *G. arboreum* and *G. hirsutum*, as well as 91 pairs of *G. arboreum* and *G. barbadense*. Among these pairs, 63 homologous gene pairs between *G. arboreum* and *G. hirsutum* displayed Ka/Ks ratios below 0.5, while two pairs had values between 0.5 and 0.99, and one pair had a ratio greater than one. Similarly, 64 homologous gene pairs between *G. arboreum* and *G. barbadense* had Ka/Ks ratios below 0.5, with two pairs having values between 0.5 and 0.99, and one pair with a ratio greater than one. In the comparison between *G. hirsutum* and *G. barbadense*, 55 homologous gene pairs exhibited Ka/Ks ratios below 0.5, six pairs had values between 0.5 and 0.99, and three pairs had ratios greater than one. Additionally, 56 homologous gene pairs between *G. barbadense* and *G. raimondii* had Ka/Ks ratios below 0.5, while five pairs had values between 0.5 and 0.99. Similarly, 59 homologous gene pairs between *G. hirsutum* and *G. raimondii* displayed Ka/Ks ratios below 0.5, with two pairs having values between 0.5 and 0.99. These results indicate that the majority of *DVL* genes have experienced strong purifying selection during their evolutionary history.

### 2.13. Transcription Analysis DVL Members

Although the sequence structure, collinearity analysis, and selection pressure analysis of the DVL gene family have been thoroughly studied, their potential role in fuzz fiber development remains unclear. We collected published transcriptome data from 12 samples (Xinluzao 50 and Xinluzao 50 FLM, and DPL971 and DPL972 from three periods) [10,16]. Xinluzao 50 and DPL971 belong to the fuzz varieties, Xinluzao 50 FLM and DPL972 belong to the fuzzless varieties. We screened the differential genes of three periods with a FPKM value greater than one, and the numbers were 10,373 and 12,318, respectively.

By combining modules with similar expressions using the dynamic shear tree method for weight values, a total of 15 modules (Figure 8A) were obtained from the materials of Xinluzao 50 FLM and Xinluzao 50. The turquoise module contained the most genes (2545 genes), while the midnight-blue module contained the least genes. With only 74 genes, each module contained an average of 691 genes.

The core modules (Figure 8B) were screened in the materials of Xinluzao 50 FLM and Xinluzao 50 according to the criteria of |r| > 0.70 and *p* < 0.0005. Among them, we found that MEred(r = 0.82, *p* < 0.000027) is highly correlated with 0 DPA, MEblue(r = 0.74, *p* < 0.00041) is highly correlated with 1 DPA, and MEturquoise(r = 0.84, *p* < 0.000016) is highly correlated with 3 DPA. MEbrown(r = 0.96, *p* < 0.00000000022) is highly correlated with 3 DPA, and it is worth noting that 3 DPA is a critical period of fuzz development. Among the 39 DVL family members of *G. hirsutum*, *GhM_A05G4176* (*GhMDVL7*) belongs to the MEred module, *GhM_A11G1425* (*GhMDVL16*) belongs to the MEblue module, *GhM_A10G2585* (*GhMDVL13*) belongs to the MEbrown module, and *GhM_A05G1032* (*GhMDVL3*) belongs to the MEturquoise module. The MEred module is mainly enriched in Ribosome, Fatty acid metabolism, Fatty acid degradation, and other pathways (Figure 8C). MEblue modules are mainly concentrated in DNA replication, spliceosome, mismatch repair, and other pathways (Figure 8D). The MEturquoise module is mainly concentrated in the ribosome, proteasome, arachidonic acid metabolism, and other pathways (Figure 8E). The MEbrown module is mainly enriched in Fatty acid elongation and the biosynthesis of secondary metabolites. In the Cutin suberine and wax biosynthesis pathways (Figure 8F), we know that fatty acids are essential for fiber development and play an important role in the elongation of cotton fiber cells. Their functional characteristics involve maintaining the structural integrity of cells and providing energy for various metabolic processes as a signal transduction medium involved in a variety of signal pathways. In addition, Cutin suberine and wax biosynthesis play an important role in the development of fuzz [21], so the genes in the MEbrown module may control the development of fuzz.

DPL971 and DPL972 obtained a total of 13 co-expression modules, with different colors representing different modules (Figure 9A). The number of genes contained in each module is different. Among them, the turquoise module contains the largest number of genes, 4102 genes, while the salmon module contains the smallest number of genes, only 75 genes, with an average of 947 genes per module.

For DPL971 and DPL972 materials, the core modules (Figure 9B) were screened according to |r| > 0.65 and ***p*** < 0.003. We found that MEblue(r = 0.95, *p* < 0.000000002) is highly correlated with 0 DPA, MEred(r = 0.7, *p* < 0.0012) is highly correlated with 3 DPA, and MEbrown(r = 0.67, *p* < 0.0023) is highly correlated with 5 DPA. Among the 20 members of the DVL gene family in *G. arboreum*, there are no members of the DVL family in the MEblue module, but the genes *Ga05G0226* (*GaDVL4*) and *Ga13G2440* (*GaDVL19*) belong to the MEred module, which is in the critical period of fuzz development. They are mainly concentrated in the MAPK-signaling-pathway-plant. Plant hormone signal transduction and signaling systems such as the Phosphatidylinositol signaling system (Figure 9C) suggest that this module may regulate the development of fuzz by influencing multiple signaling pathways. *Ga04G2089* (*GaDVL3*) belongs to the MEbrown module, which is mainly enriched in plant hormone signal transduction, the MAPK-signaling-pathway-plant signaling pathway, DNA replication, homologous recombination, and other processes (Figure 9D).

### 2.14. Functional Validation and Interaction Network Construction for GhM_A05G1032

*GhM_A05G1032*, as the target gene, is a candidate gene obtained by combining BSA-seq and RNA-seq in a previous study. qRT-PCR verified that the expression of this gene in fuzz materials and fuzzless materials had significant differences during the critical period of fuzz development. Subsequently, a VIGS experiment was conducted on this gene to see if there is an association with fuzz development. We reduced the expression of this gene in cotton, confirming the success of the VIGS experiment when the albino phenotype emerged (Figure 10A). Compared with the wild type, the expression of *GhM_A05G1032* in the three VIGS lines (line 1, line 2, and line 3) in the Xinluzao50 was decreased, and gene silencings appeared. The phenotype of the cotton was observed when the cotton was cultured further. We used empty carrier control and observed the fuzz of three different strains. We found that the fuzz of the three lines after silencing was significantly reduced compared with the fuzz of the empty carrier (Figure 10B).

Despite the preliminary validation of *GhM_A05G1032*’s function using VIGS technology, its specific interaction network remains undefined. To address this gap, we conducted a Weighted Gene Co-expression Network Analysis (WGCNA). Our analysis revealed that *GhM_A05G1032* was assigned to the MEturquoise module and, subsequently, 472 genes exhibiting a weight value exceeding 0.1 were identified as potential interactors of *GhM_A05G1032* (Appendix A). To detect the potential role of the *GhM_A05G1032* interaction network (Figure 10C), we performed KEGG analysis for these 472 genes. We found that these genes are mainly present in two basic pathways, namely protein export and homologous recombination, and some are enriched in pathways associated with fatty acid biosynthesis, which is critical for fiber development. There are also some pathways related to the secondary metabolism in the interaction network, such as glycine, serine, and threonine metabolism, biotin metabolism, and other pathways (Figure 10D).

### 2.15. qRT-PCR Analysis of DVL Gene in G. hirsutum

Our previous analysis has shown that the expression of some *GhMDVL* genes changes significantly under stressful conditions. Based on the transcriptomic data analysis and promoter *cis*-acting elements analysis, we hypothesized that five *GhMDVL* genes (*GhMDVL16*, *GhMDVL17*, *GhMDVL18*, *GhMDVL35*, and *GhMDVL38*) may be involved in resistance to biological and abiotic stress conditions. We analyzed the expression of KK1543 (drought tolerance) and Xinluzao26 (drought sensitivity), Zhongzhimian2 (disease resistance) and Simian3 (disease susceptibility), and Xinluzhong30 (salt sensitivity) and Xinluzao26 (salt tolerance) to determine whether these genes are involved in the response to stress. qRT-PCR was used to detect the transcription levels of these five *GhMDVL* genes in the root system under drought, salt stress, and *Verticillium dahliae* treatment.

Following the imposition of simulated drought treatment using PEG, the expression levels of the two tested materials exhibited significant induction at distinct time points. Notably, it is postulated that the genes *GhMDVL16* and *GhMDVL35* play a crucial role in the response of upland cotton to drought stress (Figure 11A). The gene *GhMDVL16* was significantly induced at 6 h, and its expression increased sharply, reaching the highest value, and then decreased. At 6 h, 12 h, and 48 h, there was a significant difference between the drought-resistant and drought-sensitive materials. However, the expression of *GhMDVL35* reached the highest value at 24 h, and there were significant differences between drought-resistant and drought-sensitive materials at 12 h, 24 h, and 48 h. These two genes may play a role in the response of cotton to drought stress.

In addition, after the salt stress treatment of Xinluzao26 and Xinluzhong30 with 150 mmol/L^−1^ NaCl, the genes *GhMDVL16* and *GhMDVL35* may be involved in the response of upland cotton to salt stress conditions (Figure 11B). The expression of *GhMDVL16* reached the highest value at 6 h, then reached the lowest at 12 h, and began to rise again at 24 h, and there was a significant difference between salt-tolerant and salt-sensitive materials. The expression of *GhMDVL35* reached the highest level at 6 h, and there was a significant difference between salt-tolerant and salt-sensitive materials.

Similarly, after inoculating the roots of disease-resistant and disease-susceptible materials with *V. dahlia*, the RNA transcription levels of the selected genes of the two materials at different periods were significantly induced (Figure 11C), in which the expression levels of *GhMDVL17* were significantly different at 3 h, 6 h, 12 h, and 24 h after inoculation, and the expression levels reached the highest at 12 h. *GhMDVL18* reached the highest value at 12 h after inoculation, and then began to decline, and the expression of *GhMDVL18* had significant differences at 3 h and 48 h. Similarly, the highest value of *GhMDVL38* was reached at 12 h after inoculation, and the expression of *GhMDVL38* was significantly different at 24 h and 48 h. In summary, these three genes may play a role in *V. dahlia*’s invasion of cotton.

In addition, we selected *GhMDVL14* and *GhMDVL34* for fluorescence quantitative PCR detection in the fuzz materials GZNn and fuzzless mutant GZNnFLM, as shown in Figure 11D. The expression levels of *GhMDVL14* in the two materials were significantly different 3 days after flowering. In addition, the expression of the fuzzless material was lower than that of the fuzz material. The expression level of *GhMDVL34* reached the highest on the first day after flowering, and the expression level of *GhMDVL34* was higher in the fuzzless material than in the fuzz material, with a significant difference, and then the expression level decreased. In summary, these two genes may regulate the fuzz development in *G. hirsutum*.

## 3. Discussion

The investigation of cotton fiber has gained considerable attention among scientific researchers due to its significance as a valuable cash crop in the textile industry. Understanding the underlying mechanisms of cotton fiber formation is crucial for enhancing the quality of cotton fiber. Trichomes are specialized epidermal cells found on the surface of leaves, stems, petals, and seeds in plants. The developmental pattern of trichomes in *Arabidopsis* exhibits similarities to cotton fibers [12], the study of trichoid and root hair formation may provide useful insights into the initiation and elongation of cotton fiber.

The allotetraploid cotton species, which include *G. hirsutum* and *G. barbadense*, originated from the hybridization of diploid cotton species from subgroup A and subgroup D. The availability of cotton genome data has facilitated the exploration of the evolutionary history and functional analysis of diverse gene families. The function of the DVL gene has been previously discovered in *Arabidopsis thaliana*, but no studies have been conducted on the *DVL* gene in cotton, especially in *G. hirsutum*. Previously, many gene families, including *GaGIR* [22], *GhTBL* [23], *GBSOT* [24], *GhSAC* [25], *GhLACS* [26], *GhABCG* [27], *GhKLCR1* [28], *RH2FE3* [29], *GhGSK* [30], and *GhGH3* [31] have been studied.

According to previous studies, through the analysis results of BSA-seq and RNA-seq, we screened a candidate gene (*GhMDVL3*) related to the development of fuzz, which belongs to the type of small polypeptide. According to previous studies in *Arabidopsis*, the overexpression of this gene affects the development of *Arabidopsis* trichosomes [11], which share a similar mechanism to cotton fiber development [12]. In order to verify whether this gene is related to the development of fuzz, VIGS technology was used to preliminarily verify that after the silencing of this gene, the fuzz of the three lines were significantly reduced compared with the control after silencing. We hypothesize that this particular gene may be associated with the formation of trichomes in cotton, specifically those that contribute to the development of fuzz.

### 3.1. Basic Analysis of DVL Family Genes

Based on the available reference genome data of four cotton species, a total of 117 *DVL* genes were identified through BLASTP analysis. The length of the amino acid sequences of the DVL family members ranges from 50 to 157 residues, with an average sequence length of 74 amino acids. The molecular weight of these proteins varies from 5.88 to 18 kDa, with an average of 8.59 kDa. The isoelectric point (pI) ranges from 5.89 to 11.66, with an average of 9.78, indicating that these proteins are predominantly basic in nature. Although the proteins encoded by the DVL family members exhibit diverse physicochemical properties, functions, and regulatory mechanisms, they all possess conserved DVL domains, which contribute to their structural stability.

To examine the evolutionary relationships within the DVL gene family, a phylogenetic tree was constructed. Notably, diploid *G. arboreum* and *G. raimondii* consistently clustered together with tetraploid *G. hirsutum* and *G. barbadense*, providing further evidence for the evolutionary origin of tetraploid cotton species from their diploid counterparts. Additionally, an analysis of the gene structure and conserved motifs of the DVL family members revealed that branches within the tree exhibited similar gene structures and conserved motifs. In chromosome distribution, the *DVL* gene was not evenly distributed in *G. arboreum* and *G. raimondii*. However, the distribution patterns of the two subgenomes converge, with *DVL* genes present on most chromosomes. Different from *G. arboreum*, there is no distribution of the *DVL* gene on the Chr11 chromosome, and their distribution trend is the same in tetraploid *G. hirsutum* and *G. barbadense*. The consistent heritability of these genes underscores the significance and broad functional scope of the DVL gene family [32].

Multicollinearity analysis was conducted on the DVL genes in four cotton species, namely *G. hirsutum* (AD1), *G. barbadense* (AD2), *G. arboreum* (A2), and *G. raimondii* (D5). Our findings suggest that gene amplification in the DVL family is predominantly attributed to genome-wide replication events and fragment replication events during the course of evolution. Ka/Ks ratios were calculated for 588 pairs of homologous genes across the four cotton species. The Ka/Ks ratios between *G. arboreum* and *G. raimondii* were consistently less than 0.5, as were the Ka/Ks ratios within *G. raimondii* itself. However, among tetraploid *G. hirsutum*, 49 homologous gene pairs exhibited Ka/Ks ratios less than 0.5, and three pairs had Ka/Ks values between 0.5 and 0.99. Similarly, in *G. barbadense*, 51 homologous gene pairs had Ka/Ks ratios less than 0.5, and three pairs had Ka/Ks values between 0.5 and 0.99. These observations suggest a strong purifying selection acting on these DVL genes during the course of evolution. However, one homologous gene pair between *G. hirsutum* and two homologous gene pairs between *G. barbadense* exhibited Ka/Ks ratios greater than one, indicating the positive selection and rapid evolution of these genes in recent years, potentially bearing important implications for species evolution. In summary, in the cotton DVL family, most gene pairs were purified and selected, indicating that the gene family had strong purification selection after tandem repetition, fragment repetition, and whole genome repetition. However, the selection pressure of most gene pairs was less than 0.5, indicating that *DVL* gene pairs tended to be conserved during evolution [33].

In order to better study the mechanism of *DVL* gene regulation, we also predicted the *cis*-acting elements of the *DVL* gene in *G. hirsutum* and *G. arboreum*. These include common *cis*-acting elements in promoter and enhancer regions, *cis*-acting elements involved in drought induction with MYB binding sites, and plant hormone-related *cis*-acting elements including abscisic acid response elements, salicylic acid response elements, MeJA response elements, and auxin-responsive elements, in addition to the regulatory element involved in seed-specific regulation, the light responsive element, the regulatory element involved in zein metabolism regulation, and the element involved in defense and stress responsiveness. The outcomes of our study suggest that the *DVL* gene plays a multifaceted role in various signaling pathways, as well as in plant growth, development, and defense responses. These findings provide valuable insights for the identification and selection of stress resistance genes, serving as a reference in this context [34].

### 3.2. Gene Expression and Regulation Patterns of DVL Family

Gene expression patterns are closely related to their functions. By combining transcriptome data during ovule and fiber development, we found that the expression levels of 39 *DVL* genes in the fiber development period were lower than those in the ovule development period, and the expression levels of *DVL* genes in the late ovule development period were lower than those in the early ovule development period. We hypothesized that the *DVL* gene may play an important role in the early stage of fiber development. We also investigated the changes of *DVL* gene expression in cotton pigment glands, and we found that *GhMDVL31* and *GhMDVL35* may negatively regulate the development of upland cotton pigment glands. In addition, we found that under the four abiotic stress conditions, *GhMDVL35* and *GhMDVL16* had higher expression levels than other family members, indicating that these two genes were involved in the regulation of abiotic stress. In our previous investigations, we observed that the genes *GhMDVL38* and *GhMDVL18* potentially participate in the cotton plant’s response to *Verticillium dahliae* treatment. Furthermore, the expression of *GhMDVL17* exhibited an increase at 6 h post-infection, but decreased at 12 h, indicating its potential involvement in the early response of cotton to *Verticillium dahliae* treatment. Concurrently, our findings suggest that four genes, namely *GbMDVL2*, *GbMDVL6*, *GbMDVL18*, and *GbMDVL23*, may play a significant role in conferring resistance against *Fusarium oxysporum f.* sp. *vasinfectum* (FOV) in *Gossypium barbadense*. Additionally, our investigations revealed that *GhMDVL24*, *GhMDVL2*, and *GhMDVL17* may partake in the response of cotton to thidiazuron (TDZ) treatment. According to qRT-PCR, *GhMDVL16*, *GhMDVL17*, *GhMDVL18*, *GhMDVL35*, and *GhMDVL38* may participate in the resistance to biological and abiotic stress conditions, and *GhMDVL14* and *GhMDVL34* may regulate the fuzz development of cotton.

In addition to the above studies, we conducted WGCNA analysis using the transcriptome data of *G. hirsutum* and *G. arboretum,* respectively, and found that the DVL family members were located in different types of core modules, indicating that *DVL* genes play an important role in different fiber development stages. Among 39 DVL family members of *G. hirsutum*, *GhM_A05G4176* (*GhMDVL7*) belongs to the MEred module, *GhM_A11G1425* (*GhMDVL16*) belongs to the MEblue module, and *GhM_A05G4176* (*GhMDVL7*) belongs to the MEblue module. *GhM_A10G2585* (*GhMDVL13*) belongs to the MEbrown module and *GhM_A05G1032* (*GhMDVL3*) belongs to the MEturquoise module. The MEbrown module demonstrates significant enrichment in various biological processes, including fatty acid elongation, the biosynthesis of secondary metabolites, cutin suberine biosynthesis, and wax biosynthesis, among others. We know that epidermal cells and cotton fibers need wax and keratins to stretch, and this module gene may regulate the biosynthesis of cuticle epidermal proteins, suberin, and wax, which form the outermost layer of cotton fibers.

In addition, we found that the genes *Ga05G0226* (*GaDVL4*) and *Ga13G2440* (*GaDVL19*) belong to the MEred module, which is in the critical period of fuzz development and is mainly enriched in MAPK-signaling-pathway-plant. Plant hormone signal transduction, the Phosphatidylinositol signaling system, and other signaling pathways indicate that this module may regulate the development of fuzz by affecting multiple signaling pathways.

## 4. Materials and Methods

### 4.1. Identification of Members of the Cotton DVL Gene Family

The CottonFGD database [35] was employed to access the reference genome and genome annotation data files for four cotton species: *Gossypium arboreum* (ICR), *Gossypium raimondii* (JGI), *Gossypium hirsutum* (HEBAU), and *Gossypium barbadense* (HEBAU). In order to identify the *DVL* gene in these cotton species, a Local BLASTP search was executed using the protein sequence of DVL from *Arabidopsis* as a query. Subsequently, the identified *DVL* gene was validated using the Hidden Markov Model (HMM) map retrieved from the Pfam database (PF08137) [36]. Additionally, the structure of the DVL protein domain was confirmed through an NCBI Batch-CDD search. The physical and chemical properties of the *DVL* gene were analyzed using the online website (https://web.expasy.org/compute_pi/, accessed on 3 August 2023) [37].

### 4.2. Localization of DVL Genes on Chromosomes and Analysis of Gene Duplication in Cotton

To explore the chromosomal locations of *DVL* genes in four cotton species, we obtained GFF3 files containing cotton genome annotation data from the CottonFGD database [35]. The analysis of genomic collinear blocks was conducted using MCScanX software (v1.1) [38]. Furthermore, TBtools-II software (v1.098779) was employed to analyze the physical chromosomal positions and gene duplication events of all *DVL* genes in the four cotton species, with a strict reduction in redundancy [39].

### 4.3. The Construction of a Phylogenetic Tree for the DVL Family Proteins

To elucidate the evolutionary lineage of DVL family proteins within four cotton species (*Gossypium arboreum, Gossypium hirsutum, Gossypium raimondii*, and *Gossypium barbadense*), we utilized the MEGA software (version MEGA7) and performed ClustalW multiple sequence alignments for the obtained gene sequences [40]. Utilizing the ML (maximum likelihood) method, an evolutionary tree was constructed based on the comparison results, with a Bootstrap value of 1000.

### 4.4. Analysis of Gene Structure and Conserved Protein Motifs in the DVL Gene Family

To obtain a more comprehensive understanding of the DVL family genes, we employed TBtools-II software (v1.098779) to conduct a visual analysis of the MEME file, NWK file for phylogenetic tree analysis, and the GFF3 genome annotation file of cotton [39].

### 4.5. Analysis of Expression Patterns and Cis-Elements in DVL Family Genes

The expression patterns of the *GhMDVL* gene were analyzed in various tissues, including the Root, Stem, Leaf, Petal, Receptacle, Calycle, Stamen, and Pistil, as well as in different stages of fiber and ovule development, using RNA-seq data obtained from TM-1 (PRJNA248163), sourced from the NCBI database (www.ncbi.nlm.nih.gov/, accessed on 3 August 2023) [15]. Additionally, RNA-seq data for stress conditions such as heat and cold, salt, and PEG stress were also obtained. To investigate the regulation of gene expression, the upstream 2.0 kb sequence of the start codon from DVL family genes of *Gossypium hirsutum* and *Gossypium arboreum* were extracted and used as promoter sequences for *cis*-element analysis. The plantCARE database was utilized for the further analysis of gene promoter regions and the identification of *cis*-acting elements. The obtained *cis*-acting element information was visualized using the TBtools-II software (v1.098779) [39].

### 4.6. Collinearity Analysis of DVL Family Genes

To investigate the evolutionary relationship of DVL family genes among four cotton species and identify collinear genes across the genome, we conducted a BLAST analysis using all cotton protein sequences. The MCScanX software (v1.1) was employed to compare and analyze the results of the BLAST analysis [38]. Finally, TBtools-II software (v1.098779) was used for visual analysis [39].

### 4.7. Selective Pressure Calculation

In order to study the selection pressure faced by *DVL* genes during their evolution, we used TBtools-II software (v1.098779) to calculate the non-synonymous substitution (Ka) and synonymous substitution (Ks) rates of repeating genes [39].

### 4.8. An Investigation into Gene Expression Patterns and the Construction of Weighted Gene Co-Expression Networks

For the study of *G. hirsutum*, we selected the differentially expressed genes of three different periods (0, 1, and 3 DPA) of the fuzz material Xinluzao 50 and the fuzzless material Xinluzao 50 FLM to analyze the weighted gene co-expression network. A WGCNA software (v4.1.1) package in the R program was used to construct the weighted gene co-expression network [41]. In this study, a standardized gene expression matrix was utilized as the input for investigating gene expression patterns and constructing weighted gene co-expression networks. A total of 18 transcriptome samples were obtained, representing three time points, two varieties, and three replicates each. Through threshold screening, a power value of β = 5 was selected to scale the original relation matrix, resulting in the generation of an unscaled adjacency matrix. Moreover, a minimum threshold of 30 genes per module (minModuleSize = 30) was employed. Additionally, the study focused on exploring differentially expressed genes in *G. arboreum*, using weighted gene co-expression networks based on transcriptome data obtained at different time points (0, 3, and 5 DPA) of DPL972 and DPL971.

### 4.9. KEGG Enrichment Analysis and Construction of Interaction Networks

The extraction of genes from the target module was followed by the implementation of KEGG enrichment analysis using TBtools-II software (v1.098779) [39]. The significance thresholds applied were set at *p* < 0.01 and Q < 0.05. Furthermore, to establish a potential interaction network, the Pearson correlation coefficient was computed as the interaction weight between the target genes and the candidate genes. Finally, the constructed interactive network results were visualized using Cytoscape software (v.3.7.1).

### 4.10. Cotton Material and qRT-PCR Analysis

According to our previous research results, we conducted analyses on KK1543 (drought-resistant), Xinluzao26 (drought-sensitive), Zhongzhimian2 (disease-resistant), Simian3 (disease-susceptible), Xinluzhong30 (salt-sensitive), and Xinluzao26 (salt-tolerant) materials under the conditions of inoculation with *verticillium wilt* (V991), drought stress, and salt stress. The seeds of KK1543, Xinluzao26, and Xinluzhong30 were germinated under controlled conditions of 28 °C, with a 16 h light and 8 h dark cycle, and were then transplanted into a normal hydroponic solution. The Hodgland nutrient solution was applied every two days. At the two-leaf stage, KK1543 and Xinluzao26 were treated with 15% PEG6000 for drought stress, while Xinluzao26 and Xinluzhong30 were treated with 150 mmol/L^−1^ NaCl for salt stress. Zhongzhimian2 and Simian3 were planted in soil pots, and at the two-leaf stage, the lower part of the pot was cut to damage the roots. Subsequently, a spore suspension (1 × 10^7^ spores Ml/L) of Dahlia-spore 991 spores was watered along the roots of the cotton seedlings for successful inoculation. The root tissues were collected at 0, 3, 6, 12, 24, and 48 h post-treatment. Additionally, for expression analysis, ovule and fiber samples of GZNn and GZNnFLM were collected at four different stages. The experimental setup involved conducting three biological replications and three technical replications. The real-time fluorescence quantifier 7500 was used to amplify the experiment. The analysis of gene expression was carried out using the 2^−ΔΔt^ method [42]. The primers used in this study are shown in Appendix A.

### 4.11. VIGS Silencing of Gene GhMDVL3

In order to investigate whether the *DVL* gene controls the fuzz development of *G. hirsutum*, we silenced a *GhMDVL3* gene by virus-induced gene silencing (obtained from previous work). VIGS carrier pYL156 is a laboratory storage carrier. The pYL156 vector was engineered to include the restriction endonuclease cleavage sites *SpeI* and *AscI* for the construction of pYL156: *GhMDVL3*. The specific primers for generating silenced fragments of *GhMDVL3* can be found in Appendix A. The VIGS system employed in this study consisted of four components: pYL156 (*GhMDVL3)*, pYL156 (PDS), pYL156, and pYL192.

## 5. Conclusions

This study represents the first comprehensive analysis of the DVL gene family in cotton. A total of 117 *DVL* genes were identified across four cotton species, including 20 genes in *G. arboretum* (A2), 19 genes in *G. raimondii* (D5), 39 genes in *G. hirsutum* (AD1), and 39 genes in *G. barbadense* (AD2). These genes are primarily derived from whole genome duplication (WGD) or segmental events during their evolution. Through phylogenetic tree analysis and gene structure and motif analysis, we classified the DVL family genes and predicted *cis*-acting elements. We also determined the physical locations of these genes on chromosomes, calculated the Ka/Ks ratio of homologous genes, and described the selective pressure acting on homologous genes among the four cotton species. Furthermore, the expression patterns of the DVL family genes were elucidated using RNA-seq data, and the correlation between *GhMDVL3* and fuzz development was preliminarily verified using VIGS technology. This study significantly enhances our understanding of the *DVL* gene in upland cotton and provides a foundation for further investigations into the functional roles of *DVL* genes in cotton.

## Figures and Tables

**Figure 1 ijms-25-01346-f001:**
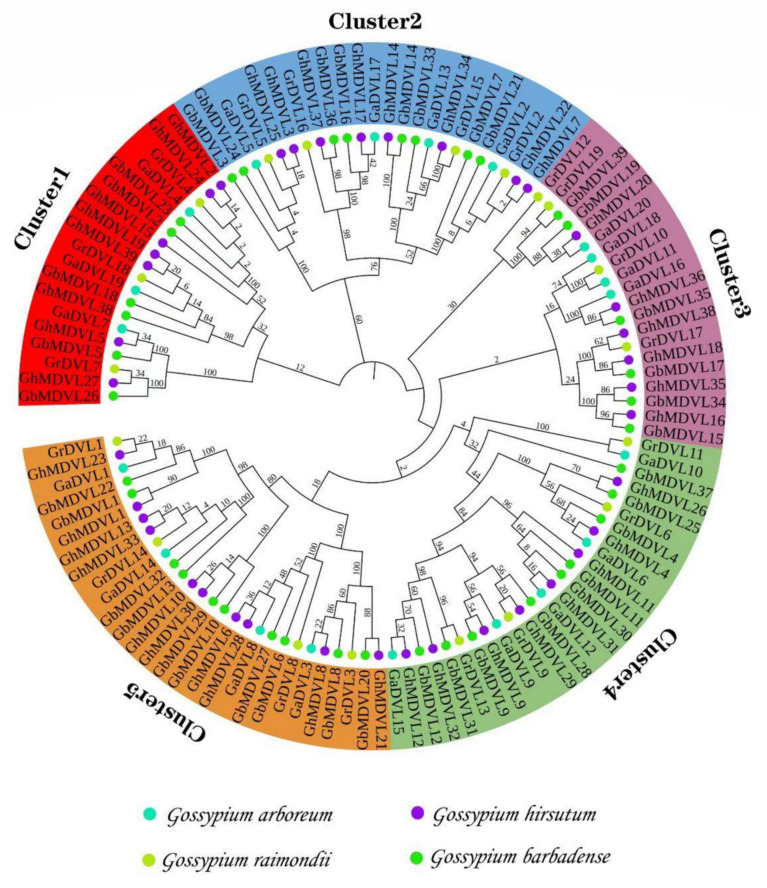
Phylogenetic analysis of DVL members in four cotton species.

**Figure 2 ijms-25-01346-f002:**
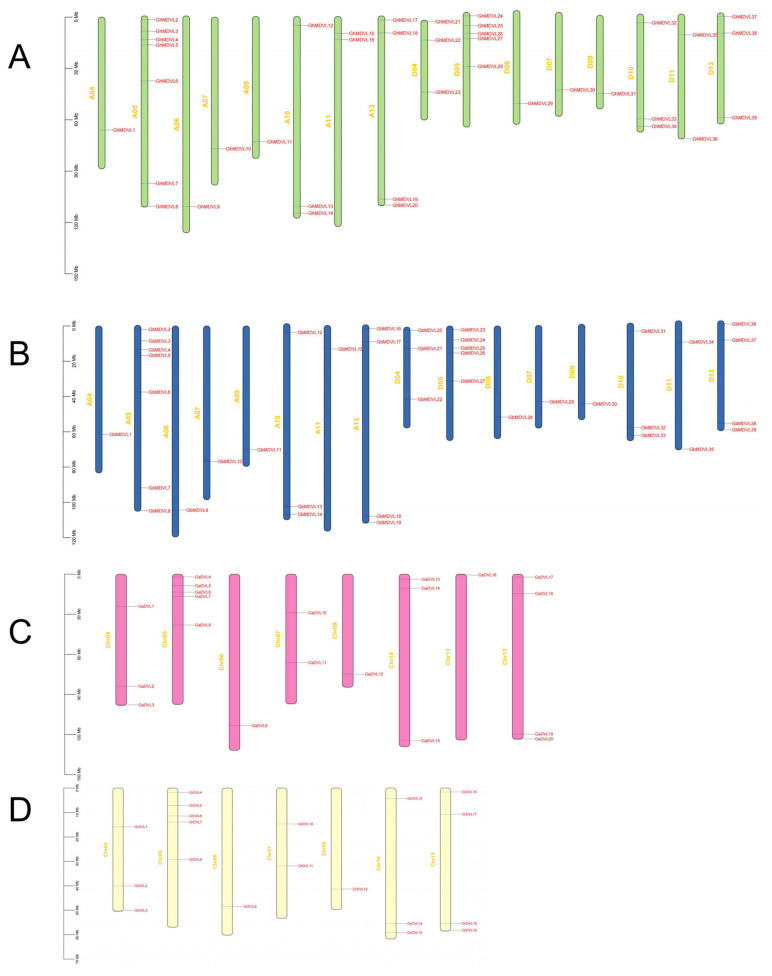
The position of the *DVL* gene on the chromosomes of four cotton species. (**A**) *G. hirsutum*. (**B**) *G. barbadense*. (**C**) *G. arboreum*. (**D**) *G. raimondii*.

**Figure 3 ijms-25-01346-f003:**
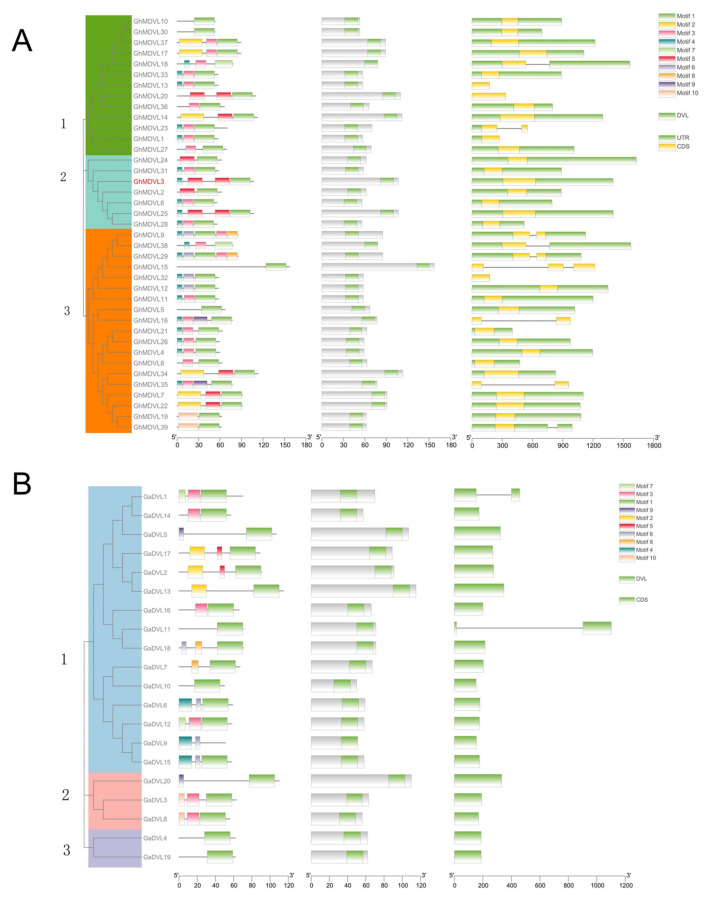
The phylogenetic tree of the *DVL* gene is divided into three groups. The motif composition and distribution, conserved domains, and gene structure (exon–intron organization) are presented from left to right. (**A**) *G. hirsutum*. (**B**) *G. arboreum*.

**Figure 4 ijms-25-01346-f004:**
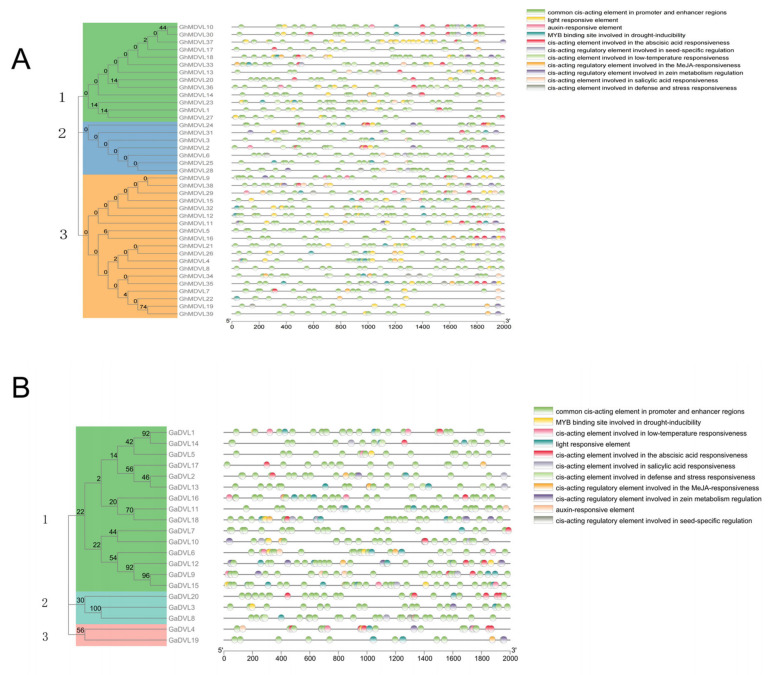
The phylogenetic tree of the *DVL* gene is divided into three groups, along with the *cis*-acting elements of the *DVL* gene promoter. (**A**) *G. hirsutum*. (**B**) *G. arboreum*.

**Figure 5 ijms-25-01346-f005:**
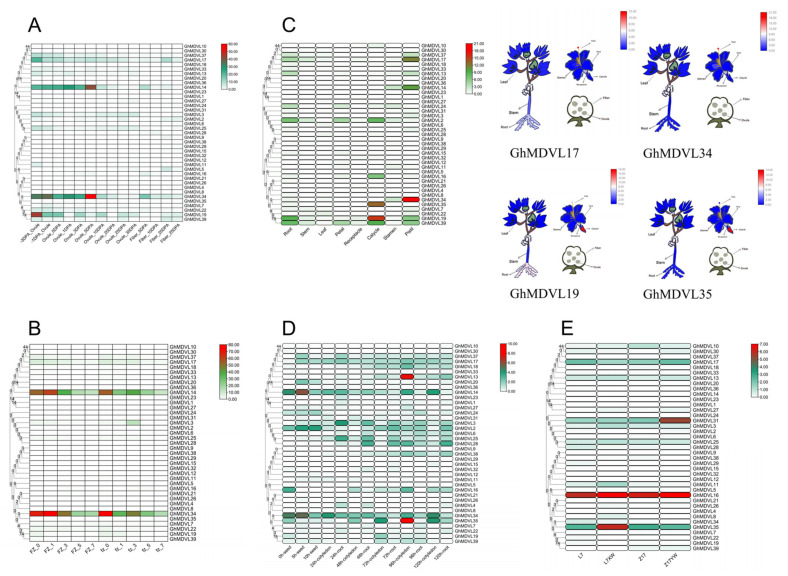
Expression patterns of *GhMDVL* genes in different tissues. (**A**) The expression profiles of *GhMDVL* genes were examined in ovule and fiber tissues at various developmental stages (−3, −1, 0, 1, 3, 5, 10, 20, 25, and 35 DPA). (**B**) The expression levels of *GhMDVL* genes were assessed in ovule and fiber tissues from fuzz material and fuzzless material at different time points (0, 1, 3, 5, and 7 DPA). (**C**) The expression patterns of *GhMDVL* genes were investigated across different organs, including root, stem, leaf, petal, receptacle, calycle, stamen, and pistil. (**D**) The expression patterns of *GhMDVL* genes were analyzed during seed, cotyledon, and root growth and development in *G. hirsutum* at various developmental stages. (**E**) The expression profiles of *GhMDVL* genes were compared between glanded and glandless materials in upland cotton.

**Figure 6 ijms-25-01346-f006:**
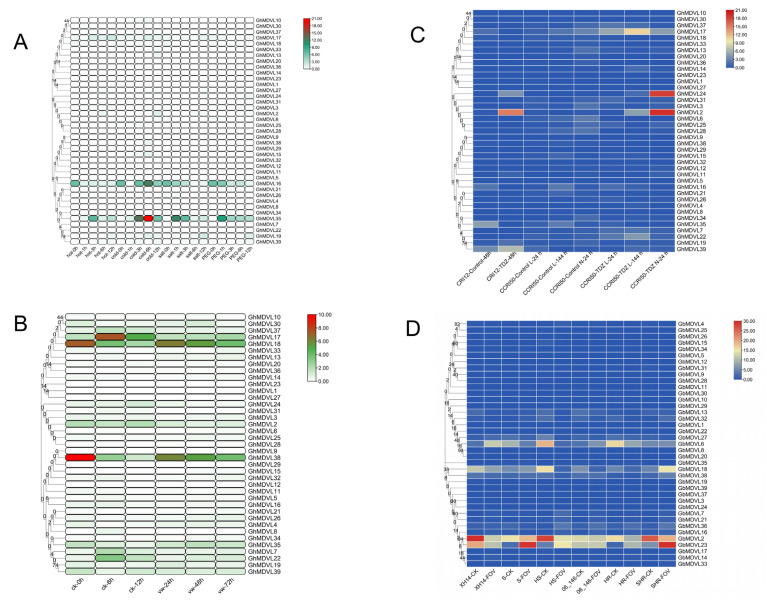
Expression patterns of *DVL* genes under different abiotic stresses. (**A**) The expression profiles of *GhMDVL* genes were examined under various abiotic stresses, including cold, hot, salt, and drought stress at different time points (0, 1, 3, 6, and 12 h). (**B**) The expression patterns of *GhMDVL* genes from *G. hirsutum* were analyzed under *V. dahliae* stress at different time points (0, 6, 12, 24, 48, and 72 h). (**C**) The expression profiles of *GhMDVL* genes from *G. hirsutum* were investigated under TDZ treatment. (**D**) The expression patterns of *GbMDVL* genes from *G. barbadense* were studied under FOV stress.

**Figure 7 ijms-25-01346-f007:**
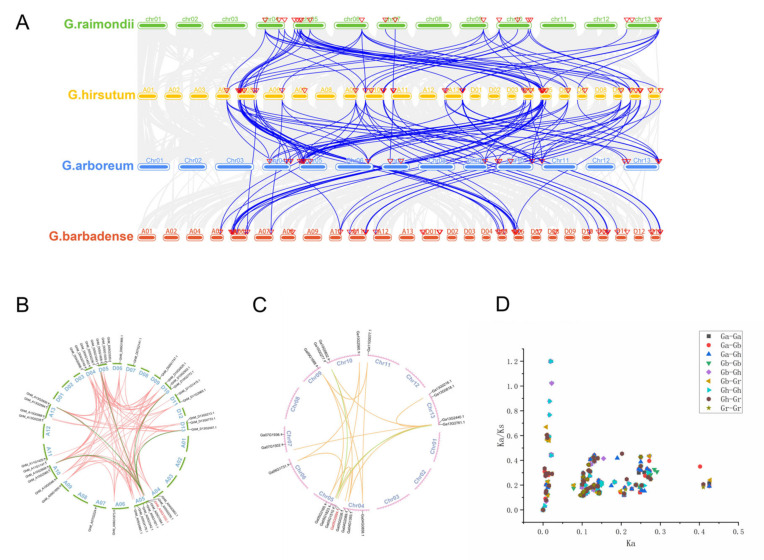
Collinearity analysis of *DVL* genes. (**A**) Multiple synteny analysis was performed to elucidate the orthologous relationship among *DVL* genes in different cotton species, namely *G. arboreum*, *G. hirsutum*, *G. raimondii*, and *G. barbadense*. The analysis employed the visualization of chromosomes of different cotton species using distinct colors. (**B**) Collinearity analysis was conducted specifically on *DVL* genes in *G. hirsutum* to investigate their structural conservation and potential evolutionary relationships. (**C**) Collinearity analysis was carried out on *DVL* genes in *G. arboreum* to evaluate their conservation patterns and potential inter-species relationships. (**D**) Selection pressure analysis was performed to assess the evolutionary dynamics and potential positive or negative selection acting on the DVL gene family during the course of evolution.

**Figure 8 ijms-25-01346-f008:**
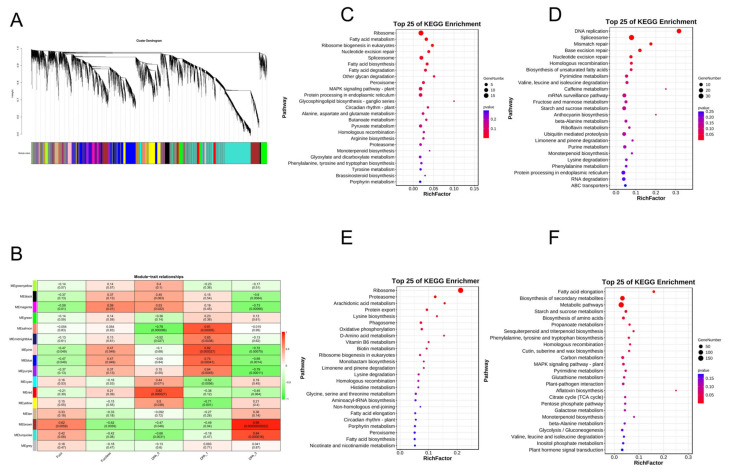
WGCNA for the transcriptome data in *G. hirsutum*. (**A**) Findings from the gene cluster analysis conducted using weighted gene co-expression network analysis (WGCNA). (**B**) Visualization of the correlation between modules and traits using a heatmap. The numbers displayed within the squares represent correlation coefficients and *p*-values between the identified modules and the traits of interest. (**C**) Assessment of the Kyoto Encyclopedia of Genes and Genomes (KEGG) pathway enrichment within the red module. (**D**) Evaluation of the KEGG pathway enrichment within the blue module. (**E**) Investigation of the KEGG pathway enrichment within the brown module. (**F**) Analysis of the KEGG pathway enrichment within the turquoise module.

**Figure 9 ijms-25-01346-f009:**
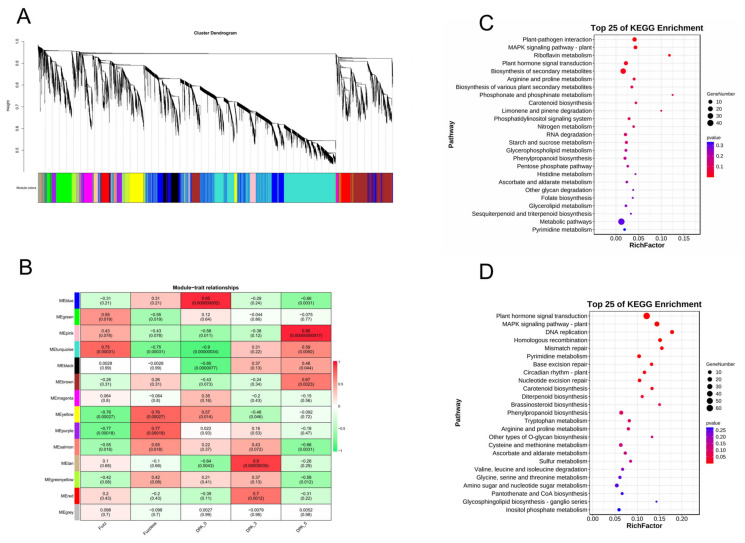
WGCNA for the transcriptome data in *G. arboreum*. (**A**) The outcomes of the gene cluster analysis conducted using weighted gene co-expression network analysis (WGCNA). (**B**) A visual representation of the correlation between different modules and traits, illustrated through a heatmap. The correlation coefficients and *p*-values between modules are denoted by numbers within the squares. (**C**) An examination of the enrichment of Kyoto Encyclopedia of Genes and Genomes (KEGG) pathways within the red module. (**D**) An investigation into the enrichment of KEGG pathways within the brown module.

**Figure 10 ijms-25-01346-f010:**
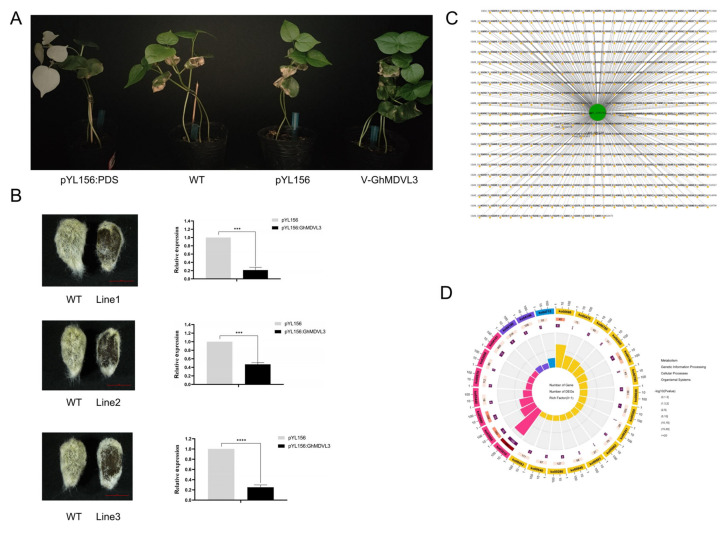
Functional verification of *GhMDVL3*. (**A**) Comparison of phenotypes of *GhMDVL3* silenced plants. (**B**) VIGS of *GhMDVL3* in *G. hirsutum*. The left images are phenotypes of fuzz on seeds after *GhMDVL3* silencing, the right images are detection of *GhMDVL3* silencing efficiency. Bars = 5 mm. (**C**) The complete network of *GhMDVL3* generated through the utilization of genome-wide transcriptome data. (**D**) An assessment of the enrichment of Kyoto Encyclopedia of Genes and Genomes (KEGG) pathways within the set of 472 genes. The error bars represent the mean values from three technical replicates ± standard errors. Statistically significant differences compared to the control group are denoted by *** *p* < 0.001, **** *p* < 0.0001.

**Figure 11 ijms-25-01346-f011:**
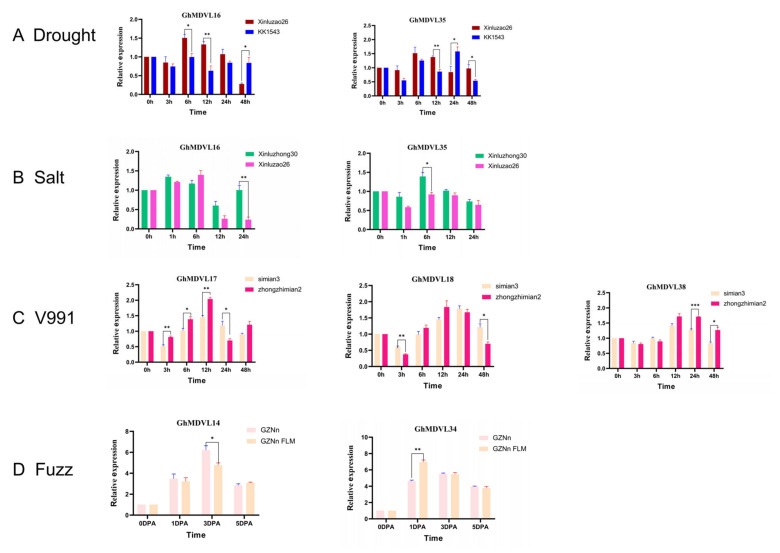
Expression profiling of the *GhMDVL* genes in *G. hirsutum*. (**A**) The transcriptional profiles of *GhMDVL* genes were examined under drought stress conditions at various time points (0, 3, 6, 12, 24, and 48 h). (**B**) The expression patterns of *GhMDVL* genes were analyzed under salt-stress conditions at different time intervals (0, 1, 6, 12, and 24 h). (**C**) The expression profiles of *GhMDVL* genes were investigated under *Verticillium wilt* conditions at multiple time points (0, 3, 6, 12, 24, and 48 h). (**D**) The transcription levels of *GhMDVL* genes were assessed during fiber development stages (0, 1, 3, and 5 DPA). The error bars represent the mean values from three technical replicates ± standard errors. Statistically significant differences compared to the control group are denoted by * *p* < 0.05, ** *p* < 0.01, and *** *p* < 0.001.

## Data Availability

The data presented in this study are available upon request from the corresponding author.

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
