# Peer review of "Genome-Wide and Expression Pattern Analysis of the DVL Gene Family Reveals GhM_A05G1032 Is Involved in Fuzz Development in G. hirsutum"

_ijms, 2024, doi:10.3390/ijms25021346_

Round 1

Reviewer 1 Report

Comments and Suggestions for Authors

This manuscript presents extensive analysis data on the DVL gene family involved in fuzz development, including bioinformatics methods (structural characterization, phylogenetic analysis), confirmation of expression patterns using RNA-seq, and identification of evolutionary relationships through in silico data mining. The vastness of the analysis data and detailed analysis present very meaningful data, and the basis is also very scientific. By providing a large amount of data, it provide readers with an in-depth understanding of the DVL gene family and information about the many genes involved in fuzz development. This is considered to be an excellent manuscript that provides readers with an understanding of not only the DVL gene but also the overall G. hirsutum gene.

All the figures from Figure 1~11, The size and resolution of the picture is so low that it is difficult to read and decipher. Please increase the resolution and image size

Author Response

Response to Reviewer 1 Comments

Dear Professor: Thank you very much for your comments on the manuscript. We have taken your suggestions in consideration point by point and made the updating of this revision.

Point 1: All the figures from Figure 1~11, The size and resolution of the picture is so low that it is difficult to read and decipher. Please increase the resolution and image size.

Response 1: Thank you so much for your suggestion.Your comments have been revised and marked in the revised version.The picture has been reuploaded.

Thanks a lot for your patience and consideration.

Sincerely yours,

Jiao Yang

Postal address: College of Agriculture, Xinjiang Agricultural University, 311 Nongda East Road, Urumqi, 830052, China.

Telephone number: 18599052306

Fax number: 0991-8762263

E-mail address:[email protected]

Reviewer 2 Report

Comments and Suggestions for Authors

-Although the manuscript title indicates the roles of DVL genes in fuzz development, some results are analyzing how these work in abiotic stress (Sections 2.7-2.10). This increases our knowledge about the gene family but dilutes the focus on development. Suggestion: Adequate the manuscript title according to the results. 

Minor concerns: 

Figure resolution: It is too low. Some figures are impossible to read (v.g. figure 3, 4, 5, and 6). 

Why are you using numbers and letters joined for mentioning time? Check the paragraph Line 228 – Line 240 (5h, 0h, 96h). There are also other words that have to be split (line 230: stage(Figure). 

Line 442: Plant hormone (no Capital letter). 

Comments on the Quality of English Language

Minor editing of English language required.

Author Response

Response to Reviewer 2 Comments

Dear Professor: Thank you very much for your comments on the manuscript. We have taken your suggestions in consideration point by point and made the updating of this revision.

Point 1: Figure resolution: It is too low. Some figures are impossible to read (v.g. figure 3, 4, 5, and 6).

Response 1: Thank you so much for your suggestion.Your comments have been revised and marked in the revised version.The picture has been reuploaded.

Point 2:  Why are you using numbers and letters joined for mentioning time? Check the paragraph Line 228 – Line 240 (5h, 0h, 96h). There are also other words that have to be split (line 230: stage(Figure).

Line 442: Plant hormone (no Capital letter).

Response 2: Thank you so much for your suggestion.Your comments have been revised and marked in the revised version.

Thanks a lot for your patience and consideration.

Sincerely yours,

Jiao Yang

Postal address: College of Agriculture, Xinjiang Agricultural University, 311 Nongda East Road, Urumqi, 830052, China.

Telephone number: 18599052306

Fax number: 0991-8762263

E-mail address:[email protected]